# Changes in Human Foetal Osteoblasts Exposed to the Random Positioning Machine and Bone Construct Tissue Engineering

**DOI:** 10.3390/ijms20061357

**Published:** 2019-03-18

**Authors:** Vivek Mann, Daniela Grimm, Thomas J Corydon, Marcus Krüger, Markus Wehland, Stefan Riwaldt, Jayashree Sahana, Sascha Kopp, Johann Bauer, Janne E. Reseland, Manfred Infanger, Aina Mari Lian, Elvis Okoro, Alamelu Sundaresan

**Affiliations:** 1Osteoimmunology and Integrative Physiology Laboratory, Department of Biology, Texas Southern University, Cleburne, Houston, TX 77004, USA; vivekmann7@gmail.com (V.M.); okoro_elvis@yahoo.com (E.O.); 2Department for Biomedicine, Aarhus University, Wilhelm Meyers Allé 4, DK-8000 Aarhus C, Denmark; dgg@biomed.au.dk (D.G.); corydon@biomed.au.dk (T.J.C.); stefan.riwaldt@med.ovgu.de (S.R.); jaysaha@biomed.au.dk (J.S.); 3Clinic for Plastic, Aesthetic and Hand Surgery, Otto von Guericke University Magdeburg, Leipziger Str. 44, 39120 Magdeburg, Germany; marcus.krueger@med.ovgu.de (M.K.); markus.wehland@med.ovgu.de (M.W.); sascha.kopp@med.ovgu.de (S.K.); manfred.infanger@med.ovgu.de (M.I.); 4Department of Ophthalmology, Aarhus University Hospital, Palle Juul-Jensens Boulevard 99, DK-8200 Aarhus N, Denmark; 5Max Planck Institute of Biochemistry, Martinsried, Am Klopferspitz 18, 82152 Planegg, Germany; jbauer@biochem.mpg.de; 6Clinical Oral Research Laboratory, Institute of Clinical Dentistry, UiO, University of Oslo, Geitmyrsveien 71 0455 Oslo, Norway; j.e.reseland@odont.uio.no (J.E.R.); a.m.lian@odont.uio.no (A.M.L.)

**Keywords:** osteoblasts, bone, tissue engineering, simulated microgravity, biomarker, cytoskeleton

## Abstract

Human cells, when exposed to both real and simulated microgravity (s-µ*g*), form 3D tissue constructs mirroring in vivo architectures (e.g., cartilage, intima constructs, cancer spheroids and others). In this study, we exposed human foetal osteoblast (hFOB 1.19) cells to a Random Positioning Machine (RPM) for 7 days and 14 days, with the purpose of investigating the effects of s-µ*g* on biological processes and to engineer 3D bone constructs. RPM exposure of the hFOB 1.19 cells induces alterations in the cytoskeleton, cell adhesion, extra cellular matrix (ECM) and the 3D multicellular spheroid (MCS) formation. In addition, after 7 days, it influences the morphological appearance of these cells, as it forces adherent cells to detach from the surface and assemble into 3D structures. The RPM-exposed hFOB 1.19 cells exhibited a differential gene expression of the following genes: transforming growth factor beta 1 (*TGFB1*, bone morphogenic protein 2 (*BMP2*), SRY-Box 9 (*SOX9*), actin beta (*ACTB*), beta tubulin (*TUBB*), vimentin (*VIM*), laminin subunit alpha 1 (*LAMA1*), collagen type 1 alpha 1 (*COL1A1*), phosphoprotein 1 (*SPP1*) and fibronectin 1 (*FN1*). RPM exposure also induced a significantly altered release of the cytokines and bone biomarkers sclerostin (SOST), osteocalcin (OC), osteoprotegerin (OPG), osteopontin (OPN), interleukin 1 beta (IL-1β) and tumour necrosis factor 1 alpha (TNF-1α). After the two-week RPM exposure, the spheroids presented a bone-specific morphology. In conclusion, culturing cells in s-µ*g* under gravitational unloading represents a novel technology for tissue-engineering of bone constructs and it can be used for investigating the mechanisms behind spaceflight-related bone loss as well as bone diseases such as osteonecrosis or bone injuries.

## 1. Introduction

Real microgravity in space can affect the health of astronauts after an extended period of time on the international space station (ISS) or other future platforms allowing a sojourn in orbit [1]. Therefore, space life-science research has been initiated, with hopes to enable humans to undertake long-term space explorations. Studies of terrestrial organisms exposed to real (r-µ*g*) or simulated microgravity (s-µ*g*) face multiple challenges, as even experiments with cells under r-µ*g* conditions are rather rare and expensive. For this reason, researchers have developed simulation devices, such as the 2D clinostat (CN), the NASA-developed rotating wall vessel (RWV) bioreactor, the random positioning machine (RPM) and the magnetic levitator, among others, to prepare for spaceflights and to conduct ground-based space research on stem cells and specialized cells [1,2,3]. RPMs are like clinostats or rotating wall vessel bioreactors, ground-based facilities constructed to simulate microgravity on the Earth’s surface (1 *g*). Their working principle is based on a gravity vector averaging to zero over time [4]. The cell samples are placed on the inner frame, where they are constantly reoriented; meanwhile, 1 *g* is always acting on the samples. The gravity vector needs to point in a specific direction for a short time period only, without acceleration of cell sedimentation. As the gravity vector averages to zero, the cells experience a state similar to microgravity. Mesland [5] proposed that the frame rotations should be faster than the investigated biological processes. Moreover, the rotation cannot be too fast, as centrifugal forces will become effective [6]. It is known that the use of an RPM induces additional forces on the cells, through the special moving pattern. It is important to mention that, when the RPM is operated within certain boundaries, these forces can be attenuated to a minimum [7]. The RPM is used worldwide for tissue-engineering purposes for various cell types and is an accepted model in preparing for future spaceflight missions [1,8].

In vitro studies on different types of human renal cortical cells or mouse MC3T3 osteoblasts in space or on microgravity simulating devices, have demonstrated significant changes in gene expression patterns [9,10], increased apoptosis (ML1 follicular thyroid cancer cells, glial cells, MDA-MB231 breast cancer cells and human lymphocytes (Jurkat)) [11,12,13,14] and induction of autophagy (human umbilical vein endothelial cells, HEK293 cells) [15,16], as well as changes in differentiation (FTC-133 follicular thyroid cancer cells) [17], migration, cell adhesion, extracellular matrix composition (ML1 cells) [11] and alterations in the cytoskeleton (FTC-133 cells, A431 epidermoid carcinoma cells) [18,19]. 

Magnetic levitation of mouse calvarial MC3T3 osteoblast cells was used as a ground-based simulation of microgravity [10]. The cells were grown on cytodex-3 beads and cultured in a superconducting magnet for 2 days, which resulted in marked alterations in gene expression. Gravitational stress leads to up- and down-regulation of hundreds of genes [10]. Random rotation and magnetic levitation induced similar changes in the actin cytoskeleton of A431 cells, which were also described in r-µ*g* [19]. 

Interestingly, it was found that tissue cells switch, in space, from a two-dimensional (2D) monolayer growth to a three-dimensional (3D) growth, into a tissue-like construct [20]. Tissue engineering in space and the application of microgravity simulation techniques is a new topic in translational regenerative medicine. Knowledge of the mechanisms of 3D growth in human cells is very important for advancing the processes of tissue engineering. Various cells exposed to the special environment of r-µ*g* and s-µ*g* conditions have already been characterized. Some examples of growing tissues from specialized cells in microgravity are: Multicellular tumour spheroids from various tumour types (MDA-MB231 and MCF-7 breast cancer cells, as well as FTC-133, ML1 and RO82-W-1 follicular thyroid cancer cells) [13,21,22,23,24,25], artificial vessel constructs (EA.hy926 endothelial cells) [26,27], regenerated cartilage (primary human chondrocytes) [28,29] or bone tissues (human pre-osteoblastic cells, human mesenchymal pre-osteoblastic cells) [30,31]. 

Tissue engineering of bone tissue is of high importance in regenerative medicine. The incidence of bone disorders worldwide is continuously increasing, due to aging populations combined with obesity and reduced physical activity [32]. The loss of skeletal tissue can accompany trauma, injury and disease. Treatment strategies include the use of stem cells, specialized cells, novel scaffolds and growth factors to improve the bone formation process [1]. Tissue-engineered bone fragments from new-born rat calvarial cells might serve as a potential alternative to the conventional use of bone grafts, as pioneered by Su et al. [33] and Hidaka et al. [34] in animal models. 

By the application of s-µ*g* methods, several preliminary studies suggested the use of osteoblast precursor or stem cells to be most suitable for the engineering of bone fragments [35]. Pre-osteoblasts, from HEPM-1460 cells, cultured in an RWV could be engineered into osseous-like tissue [30,31]. Clarke et al. designed a new method for engineering bone constructs by culturing primary osteoblasts and osteoclast precursors on a special bioreactor. This high-aspect ratio vessel (HARV) culture system provided randomized gravity vector conditions and a low-shear stress environment [36]. 

This study aims to tissue-engineer bone constructs by exposing the fast-growing foetal human osteoblasts of the hFOB 1.19 cell line—a well-characterized, stable osteoprogenitor and a widely used model for normal osteoblast differentiation [37,38]—to the RPM. Using this new method, we intend to increase our knowledge about the biology of foetal osteoblasts and to understand the effects of RPM-exposure on hFOB 1.19 cells, as indicated by differentially expressed genes and changes in protein secretion, as compared to the static 1 *g* conditions. The next step is to develop ideal bone grafts for future applications in translational studies and, subsequently, in clinical applications. 

## 2. Results

### 2.1. Morphological Examination of RPM-Exposed hFOB 1.19 Cells Reveal Signs of Organoid Formation

We detected two distinct growth patterns of hFOB 1.19 cells, exposed to simulated microgravity on an RPM for 7 days and 14 days, respectively. Human foetal osteoblasts grow in adherent monolayers (AD) during exposure to static 1 *g* conditions (for 7 days and 14 days) as shown by standard phase-contrast imaging (see Figure 1A,B). In contrast, two different hFOB 1.19 populations are distinguishable in s-µ*g*. Within the same culture flask, one group of the cells developed in the form of an adherent monolayer and the other group formed several 3D bone constructs or multicellular spheroids (MCS). These cells had detached from the monolayer and assembled into bone-like cell aggregates (see Figure 1C,D). The MCS presented an increase in size and number after 14 days of RPM exposure (see Figure 1D), as compared to 7 days RPM samples (see Figure 1C). While the bone-like constructs were observed in sizes up to 0.5 mm in length and 0.4 mm in width after 14 days, the biggest MCS was 0.2 mm in diameter after 7 days (see Figure 1F). Furthermore, using phase contrast microscopy (at magnification ×100), three times more MCS per field of view were seen on day 14, as compared to day 7 (the numbers in the inserts in Figure 1E–G). The MCS were either distributed anchorage-independently throughout the cultivation medium or found to be partially attached to the adherent monolayer. There were no MCS detectable at 7 days (Figure 1A) and 14 days (Figure 1B) in the 1 *g* control samples.

Haematoxylin–Eosin (HE)- and Sirius red-staining show the differences between the 7 days (Figure 2A,C) and 14 days RPM samples (Figure 2B,D). HE-stained MCS cross-sections are displayed at 20× (Figure 2A,B) and 10× (inserts) magnification. Numerous osteoblasts were visible in the constructs; they had produced an extracellular matrix. After 14 days, some resorption zones were detectable, where the organic matrix had broken down (Figure 2B).

Sirius red staining of the 14 days RPM samples (Figure 2D) revealed an increased deposition of collagen fibres, in contrast to the 7 days RPM samples (Figure 2C). Collagen is the most abundant organic molecule in the extracellular matrix of bone tissue [39]. Von Kossa and Alcian blue stains strongly suggested calcium deposits in the 14 days cultures (Figure 2F,H) but not after day 7 (Figure 2E,G). These stains indicate mineralization and differentiation of the cells after 14 days. 

### 2.2. Alterations of the Cytoskeleton and the Extracellular Matrix 

Changes in cell morphology depend on foregoing alterations in cytological architecture. Cytoskeletal changes are broadly acknowledged as the cause of bone loss in microgravity [40] and for organoid formation in bone tissue engineering [41]. 

Immunofluorescence was performed to elucidate the cytoskeletal changes of β-tubulin. F-actin was visualized by means of rhodamine-phalloidin staining. Well-organized F-actin filaments were observed under static 1 *g* conditions, for 7 days and 14 days of cultivation (Figure 3A,C). Following RPM exposure, a decrease in F-actin filaments was observed (Figure 3B,D). A shift in the microfilament distribution towards F-actin accumulation at the cell boundaries was clearly noticeable in both the 7 days and 14 days RPM samples. The 3D aggregates contained outer cell membranes with F-actin deposits (Figure 3B,D). MCS formation was visible. Filopodia and lamellipodia were detectable in the RPM-exposed samples. *ACTB* mRNA expression was significantly elevated in both 7 days RPM samples and further enhanced in the 14 days MCS samples (Figure 3I, Table 1). 

In accordance with F-actin, the tubulin distribution was well-organized, as expected, in the 1 *g* samples (Figure 3E). In the 7 days RPM samples, an elevated amount of tubulin protein was found in the 3D aggregates and a more intense fluorescence in the RPM-AD samples (Figure 3F).

In addition, the tubulin microtubule expression was similar in the 1 *g* controls (Figure 3G). The 14 days MCS revealed holes in the tubulin cytoskeleton of the MCS samples (Figure 3H). The *TUBB* mRNA levels were significantly increased in both 7 days RPM samples; whereas, after 14 days, *TUBB* remained significantly elevated in the RPM-AD samples (Figure 3J, Table 1). Moreover, we determined the *VIM* mRNA expression by qRT-PCR (Figure 3K), which mirrored the *TUBB* gene expression. A significant elevation of *VIM* mRNA was measured in all 7 days RPM samples; whereas, after 14 days, *VIM* was up-regulated in RPM-AD samples (Figure 3K, Table 1). Similar data were obtained by Western blot analyses of the corresponding proteins (Figure 3L–N; Table 1).

The cell adhesion and extracellular matrix protein laminin connects the cytoskeletal filaments, through transmembrane integrin receptors, with the extracellular matrix [42]. As expected, the majority of the anti-laminin positive ECM material was detectable in the cellular membranes of the hFOB 1.19 cells (Figure 4A–D). A moderate increase in laminin deposition was noticeable after 7 days RPM exposure (Figure 4B). The qRT-PCR data for *LAMA1* revealed a significant increase in both 7 days RPM samples. After 14 days, *LAMA1* gene expression did not significantly change, in comparison to the 7 days RPM samples (Figure 4E, Table 1).

The modulating influence of fibronectin on osteoblast function is well described in the literature [43]. In this study, the *FN1* gene expression was significantly elevated in adherent RPM samples at 7 days and 14 days. No change was detectable in the MCS samples (Figure 4F, Table 1).

In accordance with the morphological data, *COLA1A* gene expression was up-regulated in cells exposed to the RPM for 14 days. After 7 days RPM exposure, no significant change was detectable, compared with the 1 *g* controls (Figure 4G, Table 1). 

To investigate alterations in the ECM secretion of the hFOB 1.19 cells more deeply, we assessed the *SPP1* gene expression and the intracellular amount of protein, respectively (Figure 4H,L). The *SPP1* (i.e., the osteopontin coding gene) gene expression was significantly up-regulated in both the 7 days RPM samples and 14 days RPM samples, compared to corresponding 1 *g* controls. (Figure 4H, Table 1).

### 2.3. Changes in BMP2/TGF-β Signalling Pathways

TGF-β/BMP2 signalling has an effect on several different cellular processes. Their important functions for osteogenesis in mammals are widely accepted [44]. TGF-β_1_ promotes bone matrix synthesis, as well as differentiation of osteoblast progenitors [44]. 

The *TGFB1* gene expression was not significantly changed but showed a tendency to increase after 7 days. It was significantly up-regulated in the 14 days RPM-AD samples (Figure 5A, Table 1). The TGF-β_1_ protein was elevated in both AD and MCS after 7 days but not elevated in MCS after 14 days (Figure 5E, Table 1). 

The *BMP2* expression was contrarily regulated. We detected a significantly decreased *BMP2* gene expression level in both the 7 days and 14 days RPM-AD samples, compared to the corresponding 1 *g* controls. The *BMP2* expression was clearly up-regulated in the 7 days MCS (Figure 5B, Table 1). The BMP2 protein levels showed a counter-regulatory behaviour (Figure 5F).

As published earlier, hFOB 1.19 cells are able to differentiate to a chondrogenic state [45]. *SOX9* is an acknowledged marker for proliferating chondrocytes and it is recognized as a crucial transcription factor for differentiation of precursor cells into chondrocytes [46]. Our studies revealed a significant increase in *SOX9* expression levels of RPM-exposed adherent foetal osteoblasts in both 7 days and 14 days-experiments (Figure 5C, Table 1). In addition, we measured the mRNA of the proliferation marker Ki-67 (*MKI67*) in all groups but did not detect an altered gene expression (Figure 5D, Table 1). However, the Ki-67 antigen varied and was three-fold higher expressed in the constructs, compared to the control cells, after 2 weeks of culturing on the RPM (Figure 5H, Table 1). 

It is known that BMPs influence Wnt signalling in bone remodelling through Dkk1 and SOST [44]. A negatively regulated bone mass, through increased osteoclastogenesis caused by increased RANKL-signalling, due to reduced Wnt-signalling by increased Sost concentrations, has already been elucidated in an animal model [47].

### 2.4. Secretion of Soluble Factors in the Supernatant

Differentiation of osteoblast precursor cells to osteoblasts is inhibited by TNF-α [48]. Additionally, TNF-α acts as an inflammatory cytokine in several skeletal diseases [49]. In our study, the TNF-α secretion was reduced in 7 days RPM samples but remained stable after 14 days (Figure 6A, Table 1).

Besides TNF-α, IL-1β is known as a classical mediator of acute inflammation. In general, IL-1β is known to attenuate bone formation and reduce osteoblastogenesis [50]. IL-1β secretion was significantly decreased in the supernatant of osteoblasts exposed for 7 days to the RPM, compared to the 1 *g* controls. In contrast, a 14 days RPM exposure induced the release of IL-1β, as compared to the 14 days 1 *g* controls (Figure 6B, Table 1).

In addition, osteopontin secretion in the supernatants was significantly up-regulated in the supernatant of RPM samples after both 7 days and 14 days, compared to corresponding controls (Figure 6C, Table 1). Osteocalcin (OC), which, in particular, is synthesized by osteoblasts, maintains a central role in bio-mineralization [51]. OC concentrations of the secreted proteins were significantly elevated in the 14 days RPM samples, as compared with the corresponding control (Figure 6D, Table 1).

The release of Dkk1 remained unchanged in all study groups (Figure 6E, Table 1), whereas the SOST concentration was significantly reduced in the 7 days RPM samples (Figure 6F, Table 1); after 14 days, a lower amount of SOST was released by the osteoblasts in the supernatant. A 14 days RPM exposure did not change the release of SOST by the cells (Figure 6F, Table 1). 

Osteoprotegerin (OPG) skews the balance between osteoclastogenesis and osteoblastogenesis towards increased bone formation by decreasing RANK activation. The RANKL decoy receptor, OPG, is a Wnt-pathway target gene. Wnt-activation leads to up-regulated OPG synthesis [52]. In the 7 days experiment, OPG secretion was significantly reduced by RPM exposure; whereas, after 14 days, a clear elevated OPG release in RPM-exposed cultures was detectable (Figure 6G, Table 1). The HBNMAG-51K (MILLIPLEX MAP) bone metabolism multiplex assay also investigated ACTH, IL-6, insulin, leptin, PTH and FGF-23. None of the six proteins were released in detectable amounts by the foetal osteoblasts.

### 2.5. Interactions of Proteins Involved in Bone Formation

We employed Pathway studio v11 to probe the linkage between the proteins detected experimentally, either in the supernatant (by Luminex technology) or within the cells (measured by Western blotting). Figure 7A represents the interaction analysis as a network of interacting proteins, such as the bone markers OPG (TNFRSF11B). In vivo OPG binds RANKL and blocks its interaction with RANK [53]. The binding may be inhibited by collagen 1α1 [54], while TNF-α enhances the concentration of both proteins [55]. In addition, sclerostin (SOST) and the bone morphogenic protein 2 (BMP2) have an influence on the RANKL/OPG system. However, SOST inhibits vimentin and β-actin through DKK1 [56], while elevating the expression of *SOX9* [57]. Additional central points of the overall network are laminin α, fibronectin and TGF-β_1_ [58]; as well as osteopontin, which interacts with vimentin [59] and fibronectin [60]. Figure 7B presents a mutual interaction of the expression of the genes, investigated by qRT-PCR and shown in Figure 3I–K, Figure 4E–H, Figure 5A–D and in Table 1. 

## 3. Discussion 

In the present study, we investigated human foetal osteoblasts (hFOB 1.19 cell line), either exposed to the RPM or cultivated under static 1 *g* conditions for 7 days and 14 days, respectively. 

The objective of this study was to examine the potential of the osteoblast constructs as suitable biomaterial to support human osteogenesis, with potential application in bone tissue engineering. Manifold attempts to engineer tissue substitutes by exposing cells to simulated microgravity, using different methods, such as the RPM, RWV, 2D-clinostat or magnetic levitation, have been made [1,24,27,28,29,30,31,32,61,62,63]. These techniques had been applied to culture multicellular cancer spheroids from thyroid cancer cells (FTC-133 [17,25], ML-1 [11,24] and RO82-W-1 [24] cells), breast cancer cells (MDA-MB-231 [13,61,63] or MCF7 [22,23,64]), cartilage tissue pieces (human chondrocytes [28,29], primary sheep chondrocytes [62]) and 3D aggregates from stem cells [1,61,63], endothelial cells [26,27,61] and others [61]. These new methods offer new applications in tissue engineering and 3D bioprinting, are important for drug targeting and discovery and show potential in the field of cancer research [1,20,35].

Practical applications need further improvement, however, which may be achieved by discovering possibilities for intervening the process, as we have shown for the cells of the MCF7 cell line [23,64]. Therefore, we investigated the underlying mechanisms involved in the development of the spheroids. We focused on cytoskeletal changes (F-actin microfilaments and tubulin microtubules), the ECM and the growth and multicellular spheroid formation of the hFOB 1.19 cells when subjected to the RPM or to 1 *g* conditions. In addition, we studied the expression of genes coding for proteins involved in cell structure, shape, migration, adhesion or angiogenesis and hence which might play a role in the cellular capacity to sense gravitational alterations and the development of MCS, demonstrated with FTC-133 follicular thyroid cancer cells or MCF-7 breast cancer cells [17,21,22]. 

Interestingly, we observed that a minor fraction of the hFOB 1.19 cells grew in the form of 3D constructs or MCS after 7 days RPM exposure. 

Applying the experimental design, described in the material and methods section, which was based on recent experience with culturing the FTC-133 thyroid and MCF7 breast cancer cells, as well as endothelial cells under real and simulated microgravity [17,21,22,26,27], we repeatedly detected that hFOB 1.19 cells developed into two phenotypically different populations; one subpopulation remaining adherent to the bottom of the culture flask (the RPM-AD monolayer) and another subpopulation that formed MCS, freely floating in the cultivation medium. The number and size of the MCS increased after 14 days RPM exposure (Figure 1), suggesting that the reduced gravity conditions provided by the RPM promoted the development of MCS and 3D growth, even if no scaffold was present—as demonstrated earlier with the help of an RCC, using human embryonic palatal mesenchymal pre-osteoblasts [65]. The MCS contained numerous cells and ECM deposition was evident at both 7 days and 14 days, along the surface of the tissue-engineered construct. The construct (MCS) size and number changed with time, particularly between 7 days and 14 days (see Figure 1). Constructs appeared to be confluent with cells at 14 days, both at the periphery and centre of the construct. As demonstrated in Figure 2, an accumulation of ECM proteins was detectable with time, as well as calcification areas, which were visible after 14 days (as demonstrated by the von Kossa and Alcian blue stains). 

F-actin staining revealed increases in F-actin at the cellular membranes of the MCS after 7 days and 14 days. Actin, tubulin and vimentin are key components of the cytoskeleton and have various functions [66,67]. They play an important role in the perception of exterior signals, including the gravitational force [66]. Notably, cytoskeletal alterations have been detected in cell lines derived from various cell types, such as lymphocytes, glial cells, endothelial cells, cancer cells and others [11,12,14,21,22,23,26,28]. Unfortunately, studies investigating the cytoskeleton in r-µ*g* and s-µ*g*, so far, are based on the analysis of fixed cells. Live cell imaging of FTC-133 cancer cells expressing the Lifeact-GFP marker protein for the visualization of F-actin under real microgravity during a TEXUS sounding rocket mission and a parabolic flight mission using a compact fluorescence microscope (FLUMIAS) confirmed the results obtained with fixed cells [18]. 

During the TEXUS rocket flight, well-structured filament bundles were found in FTC-133 thyroid cells, as well as stress fibres, together with filopodia- and lamellipodia-like structures and cell detachment [18]. These experiments, performed in s-µ*g* and r-µ*g*, revealed clear changes in morphology, cytoskeleton and function. Studies on microtubules in altered gravity conditions have shown that they are also gravity-sensitive [67]. 

In this study, we detected similar cytoskeletal changes of F-actin at the outer cellular membrane after both 7 days and 14 days (see Figure 3). Moreover, we found up-regulated *ACTB*, *TUBB and VIM* in the AD and MCS samples after 7 days RPM exposure. *VIM* was significantly down-regulated after 14 days in the RPM-AD samples. The *TUBB* gene expression level of the 1 *g* samples was similar to that of the MCS group (Figure 3).

The 1 *g* control samples, grown in T25 culture flasks, multiplied as a monolayer and never formed any MCS. In contrast, hFOB 1.19 cells exposed to the RPM presented 3D growth. Histological investigation of these constructs showed both round and irregularly-formed structures, with a uniform cellular distribution throughout the constructs at 7 days and 14 days (Figure 1 and Figure 2). An external layer (or capsule) of cells developed on the surface of the constructs, becoming more prominent with time. There was no histological difference between 7 days or 14 days-RPM exposure. Sirius red staining revealed an ECM deposition in the MCS, detectable at both 7 days and 14 days. The ECM was composed of ECM proteins secreted by the cells, thereby providing important structural, mechanical and biochemical support to the surrounding cells [68]. We observed several alterations in the components of the ECM. The gene expression of *SPP1* was clearly up-regulated in all RPM samples. The *FN1* mRNA expression was significantly elevated in AD cells, compared to 1 *g*, whereas no changes were found in MCS. Collagen type 1 is a marker for early stage osteogenesis, which was expressed in all cultured constructs. The *COL1A1* mRNA was not significantly altered in the 7 days samples but was significantly elevated in the 14 days samples. In contrast, *LAMA1* was significantly up-regulated in all 7 days RPM samples but was not altered in the 14 days samples. 

In accordance with these findings, CLSM analysis revealed an apparent increase in laminin in the cells exposed to the RPM, compared to the 1 *g* controls. Even though CLSM is not a quantitative method, the obtained images indicated an overall increase in anti-laminin-positive signals in the RPM samples, compared with the 1 g controls. Moreover, the CLSM images also demonstrated that F-actin and laminin were accumulated at the cell boundaries and in the extracellular space around the MCS, which corresponds well to earlier findings on different types of cells such as glial cells, lymphocytes (Jurkat) and FTC-133 thyroid cancer cells [12,14,18].

Bone morphogenetic protein 2 (BMP-2) belongs to the TGF-β superfamily; it is involved in cartilage and bone development and has demonstrated an osteo-inductive effect [69,70]. Patients receiving exogenous BMP-2 to treat bone fractures or defects exhibited elevated sera levels of TNF-α and IL-1β, which may affect the outcome of bone regeneration [50]. *BMP2* gene expression was significantly reduced in RPM AD cells after both 7 days and 14 days. In 7 days samples, BMP2 was elevated in MCS, compared to 1 *g*, indicating BMP2 playing a role in early bone formation, in this experimental setting. 

Transforming Growth Factor β1 (TGF-β1) has been shown to influence cell attachment of bone cells of the line MG-63 [71]. TGF-β1 is involved in the regulation of cell proliferation, differentiation and growth [44]. In addition, it is linked with embryogenesis, tissue repair and bone regulation [44]. 

There was a trend of increased *TGFB1* mRNA levels in the 7 days RPM samples. We measured a significantly elevated *TGFB1* mRNA level in the 14 days RPM-AD cells, compared to the 1 *g* samples. Pathway analyses (Figure 7) revealed interactions between TGFB1 and LAMA1, as well as between TGFB1 and FN1. TGF-β_1_ is known to enhance ECM deposition in several tissues. 

The SRY HMG box-related gene 9 (*SOX9*), which plays an important role in normal skeletal development, was up-regulated in both the 7 days- and 14 days-RPM-AD cell samples, as compared to the 1 *g* samples. No change was found in the MCS. The BMP2, TGFB1 and SOX9 proteins are involved in chondrogenic and osteogenic differentiation [72]. 

In summary, the mRNAs of *BMP2*, *TGFB1* and *SOX9* were affected in foetal osteoblasts exposed to s-µ*g*. This was similar to earlier studies, which demonstrated the positive influence of these factors on human chondrocytes exposed to parabolic flight manoeuvres [73]. The relevant proteins, such as BMP-2, represent growth and differentiation factors regulating the chondrocyte development through a complex mutual regulation [74]. The products of TGFB1 and SOX9, together with BMP-2, drive development from mesenchymal stem cells toward cartilage-forming chondrocytes [75]. This progress appears to be supported, because BMP-2, TGFB1 and SOX9 gene expression and protein content were enhanced after 31 parabolic manoeuvres [73]. 

We also focused on the *MKI67* mRNA expression, which was not significantly altered in all samples, while the Ki-67 antigen was clearly enhanced in the 14 days RPM-MCS cell samples (Figure 5H) indicating that this group had an enhanced proliferation behaviour in the experimental setting.

We further investigated the secretion of the cytokines involved in bone formation from hFOB cells. The release of dickkopf-related protein 1 (DKK1) in the cell supernatant was unchanged after 7 days and 14 days, between both control and RPM samples but the hFOB cells revealed a time-dependent reduction of the protein after 14 days, compared to 7 days. DKK1 is a secreted Wnt inhibitor that binds LRP5 and LRP6 during embryonic development. There is evidence that a reduction in DKK1 elevates Wnt activity and induces a high bone mass phenotype. Reduced levels of DKK1 remove suppression of bone formation and prevent the development of osteolytic bone diseases. Targeting DKK1 might have therapeutic significance for the treatment of low bone-mass diseases [76]. 

Furthermore, a key signalling factor controlling bone resorption and bone formation is the receptor activator of nuclear factor-κB (RANK)/RANK ligand/osteoprotegerin [77]. Osteoprotegerin (OPG) is a cytokine receptor, which also belongs to the tumour necrosis factor (TNF) receptor superfamily. 

Pro-inflammatory cytokines, extracellular matrix proteins and bone-regulating cytokines of the RANKL/OPG and Wnt/β-catenin pathways are mandatory for normal bone repair [78].

The OPG level was reduced after 7 days in RPM samples, as compared to the controls. In contrast, after 14 days, the secretion of OPG was significantly elevated and further enhanced by RPM exposure. 

Osteocalcin (OC), also called bone gamma-carboxyglutamic acid-containing protein (BGLAP), is synthesized by osteoblasts and is involved in the metabolic regulation. OC acts as a pro-osteoblastic (anabolic) factor and is important for bone formation. Furthermore, it has also been implicated in bone mineralization and calcium ion homeostasis [79]. 

OC secretion remained stable after a 7 days exposure to the RPM and in the 1 *g* control osteoblasts. In contrast, the release of OC was significantly elevated in 14 days RPM-exposed osteoblasts, as compared to static controls. These results fit well to the finding that, after 14 days, the bone tissue constructs were increased in both number and size. The pathway analysis revealed that OC interacted with *SPP1*.

Osteopontin is also known as secreted phosphoprotein 1 (SPP1). It is an extracellular structural protein and, therefore, an organic bone component. It is also detectable in various cell types (e.g., FTC-133 follicular thyroid cancer cells and EA.hy926 endothelial cells) and has been shown to be altered by microgravity conditions [25,27,80]. 

We measured the osteopontin (*SPP1*) gene expression and found that *SPP1* mRNA was significantly elevated after both 7 days and 14 days in the RPM samples, as compared to the static controls. This finding is in agreement with data obtained from thyroid cancer cells and endothelial cells, which both showed elevated *SPP1* mRNAs after RPM exposure [25,27,80]. 

Sclerostin (SOST) is produced by osteocytes and has anti-anabolic effects on bone formation; inhibition of SOST can stimulate bone formation [81]. The SOST secretion was significantly reduced after 7 days in the RPM samples, as compared to the 1 *g* controls. 

After 14 days, the cells secreted unaltered levels of SOST in the supernatant. This indicates that, initially, microgravity inhibits SOST and stimulates bone formation, but, after 14 days, microgravity did not support bone formation through this mechanism any longer.

Pro-inflammatory cytokines are important for bone formation and bone repair [78] and, therefore, we measured the secretions of IL-1β and TNF-α in the supernatant.

The cytokine interleukin-1β (IL-1β) is an important mediator of the inflammatory response and is involved in several biological processes, such as proliferation, differentiation and apoptosis [82]. The induction of cyclooxygenase-2 (PTGS2/COX2) by this cytokine in the central nervous system is found to contribute to inflammatory pain hypersensitivity [83]. The release of IL-1β decreased significantly after 7 days in RPM samples, as compared to the 1 *g* controls. After 14 days, foetal osteoblasts in the RPM samples released increased amounts of the IL-1β protein, as compared to the 1 *g* controls. The amount of released IL-1β was significantly higher after 14 days, compared to the amounts after 7 days.

Furthermore, we investigated the release of tumour necrosis factor alpha (TNF-α), which is a cytokine involved in the regulation of a wide spectrum of biological processes, including cell proliferation, differentiation, apoptosis, lipid metabolism and coagulation [84]. 

The release of TNF-α decreased after 7 days in RPM samples, as compared to the static 1 *g* controls. After 14 days, the hFOB cells of both groups only secreted a low and unaltered amount of TNF-α in the supernatant.

Taken together, these in vitro tissue-engineered bone constructs provide an accurate system for investigating the molecular signaling pathways involved in normal bone formation and remodeling processes and could be also used for co-cultures with osteoclasts. Additionally, as an in vitro system, it has the potential of reducing animal experiments, for instance as an alternative method for testing the effects of pharmaceutical agents or biochemical stimuli. Moreover, these bone constructs can be used for investigating bone-related diseases. Patients having plastic and/or reconstructive surgery with traumatic bone defects or osteonecrosis could benefit from this approach in the future.

## 4. Materials and Methods

### 4.1. Cell Culture

The cell culture model used was represented by a human foetal osteoblast cell line, hFOB 1.19 (catalogue number CRL-11372; American Type Culture Collection, Manassas, VA, USA), which is a conditionally-immortalized clonal cell line [37]. hFOB 1.19 is immortalized with a gene encoding for a temperature-sensitive mutant of the SV40 large T antigen (tsA58). 

We cultured hFOB 1.19 in a 1:1 mixture of DMEM and Ham’s F12 media (Lonza, Verviers, Belgium) supplemented with 2.5 mM l-glutamine (without phenol red), 10% foetal bovine serum (Biochrom AG, Berlin, Germany) and 0.3 mg/mL G418 (Gibco, Life Technologies, Naerum, Denmark). The cells were seeded in 75 cm^2^ ventilated cell culture flasks (Sarstedt AG, Nümbrecht, Germany) under the standard conditions of 37 °C and 5% CO_2_, as previously described [21,22,23]. 

Cells in the third to fifth passage were used in the subsequent experiments. For RPM experiments, hFOB 1.19 cells were seeded into 25 cm^2^ ventilated cell culture flasks (Sarstedt AG) and given at least 24 h to properly attach to the bottom of the flasks, forming an adherent monolayer (AD). Subsequently, the flasks were completely filled with the supplemented medium. The experiment ended with the fixation of cells assigned to s-µ*g* or controls, as described earlier [21,22,23]. For 14 days experiments, a 50% medium exchange by aspiration was necessary after 7 days, to ensure proper concentrations of nutrients and waste products. The remaining MCS were retrieved by upright positioning of flasks, leading to MCS sedimentation.

### 4.2. Random Positioning Machine

Microgravity conditions were simulated using a desktop RPM (Airbus Defence and Space, former Dutch Space, Leiden, The Netherlands) [2]. The RPM rotates a central frame around two perpendicular axes, randomly changing the direction of rotation and the angular velocity, in the range between 60°/s and 75°/s, in both axes (Figure 8). It was used in the real random mode. Thereby, the direction of the gravitational acceleration affecting the samples was continuously randomized and the magnitude of the net gravitational vector approached zero over time, producing a simulated state of microgravity [4,7]. The RPM was placed inside a commercial incubator, under standard conditions at 37°C and 5% CO_2_, with up to twelve 25 cm^2^ flasks fixed to the central frame. The flasks were placed no further than 7.5 cm from the centre of rotation. 

All 25 cm^2^ flasks were completely filled with medium, minimizing shear stress. The samples were run on the RPM for 7 days and 14 days, while an equivalent number of 25 cm^2^ flasks, also completely filled with medium, were placed in the same incubator, to serve as 1 *g* controls to the RPM samples [21,22,23]. 

### 4.3. F-Actin, Tubulin and Laminin Staining

For F-actin, laminin and tubulin staining, 5 × 10^5^ cells were seeded in 60 Nunc™ LAB-TEK™ slide flasks (nr. 170920, material: polystyrene, Fisher Scientific, Roskilde, Denmark). 

The attached cells were cultured for 7 days or 14 days, respectively, on the RPM (*n* = 30) or next to the RPM, acting as 1 g controls (*n* = 30) in the same incubator. The slide flasks were filled completely and made air bubble free with medium. 

For the 14 days experiment, 50% of the medium in the cultivation flasks was carefully aspirated and exchanged with fresh medium after 7 days. After 7 days or 14 days of cultivation, the cells were washed twice with DPBS (Life Technologies™, Carlsbad, CA, USA) and fixed, with 4% paraformaldehyde and 70% methanol, for 30 min. After fixation, the cells were permeabilized with a Triton-X-100 (Merck, Darmstadt, Germany) and, subsequently, washed twice with DPBS. 

Laminin and tubulin were visualized by immunofluorescence, using primary antibodies against laminin (1:25, L9393, Merck) and tubulin (1:500, T5293, Merck). 

After the primary antibody incubation for 24 hours at room temperature, the cells were washed twice with DPBS. Subsequently, the samples were incubated with anti-rabbit (laminin) or anti-mouse (tubulin) secondary antibodies, conjugated to an Alexa Fluor 488 (1:500, 4412S and 4408S, both Cell Signalling Technology, Danvers, MA, USA). 

F-actin was visualized by means of rhodamine-phalloidin staining (Molecular Probes^®^, Eugene, OR, USA). For the detailed method, see [26]. Fluoro-shielding with 4′, 6-diamidino-2-phenylindole, DAPI, (Sigma) was used for nuclear counterstaining and mounting in all samples, before subsequent imaging by confocal laser scanning microscopy (CLSM) [26,85]. 

### 4.4. Microscopy

Phase contrast microscopy was performed before and after the RPM experiments, to ensure viability and determine the morphological changes of the cells. Pictures were taken using a Canon EOS 550D camera (Canon GmbH, Krefeld, Germany) through a Leica DM IL LED inverted microscope (Leica Microsystems, Wetzlar, Germany). Fluorescence staining was analysed using a Zeiss LSM 710 CLSM (Zeiss, Jena, Germany) fitted with a Plan-Apochromat 63 × 1.4 objective, as previously described [26,85]. Excitation and emission wavelengths for the Alexa Fluor 488 were λ_ex_ = 488 nm and λ_em_ = 525 nm, respectively. Correspondingly, for the TRITC: λ_ex_ = 532 nm and λ_em_ = 576 nm.

### 4.5. RNA Isolation

The cells were removed from the culture flasks with scrapers and transferred immediately to tubes for isolation by centrifugation at 3400× *g* for 10 min at 4 °C. Total RNA was extracted using an AllPrep RNA/Protein Kit (Qiagen GmbH, Hilden, Germany), according to the manufacturer’s instructions. RNA concentrations and quality were determined spectrophotometrically at 260 nm with a SpectraMax M2 Microplate Reader (Molecular Devices, Sunnyvale, CA, USA). 

The isolated RNA had an A260/A280 ratio of >1.5. The cDNA for quantitative PCR was obtained using a First-Strand cDNA Synthesis Kit (Fisher Scientific), according to the manufacturer’s instructions.

### 4.6. Quantitative Real Time PCR

Quantitative real-time PCR (qRT-PCR) was used to determine the expression levels of target genes [21,22,23,26]. Appropriate primers (overview given in Table 2) were designed using the Primer express software (Applied Biosystems, Darmstadt, Germany) with a T_m_ of 60°C, taking care of the exon-exon boundaries. All primers were synthesized by TIB Molbiol (Berlin, Germany). The cDNA analysis was performed using a 7500 Fast Real-Time PCR System with Fast SYBR Master Mix (both from Applied Biosystems). The final reaction volume was 12.5 µL, including 1 µL template cDNA and a primer concentration of 500 nM. PCR conditions were as follows: 20 s at 95 °C, 40 cycles of 3 s at 95 °C, 1 min at 60 °C, followed by a melting curve analysis, with a temperature gradient from 60 °C to 95 °C in 0.3 °C increments. If all amplicons showed a T_m_ similar to the one predicted by the Primer Express software, the PCR reaction was considered to be specific. Each sample was measured in triplicate. Relative transcription levels were quantified using the comparative C_T_ (ΔΔC_T_) method. The transcription level of the housekeeping gene *18S rRNA* was used as a reference gene for data normalization.

All samples were measured in triplicate and normalized to the housekeepers 18S mRNA and TBP. The 7 days 1 *g* samples were set as 100%. 

### 4.7. Determination of Intracellular Protein Levels

Western blot analyses, including gel electrophoresis, trans-blotting and densitometry, were performed following routine protocols, as described previously [64,86]. Primary antibodies for protein detection and used dilutions are listed in Table 3. All samples were normalized to the housekeeper protein cyclophilin.

### 4.8. Measurement of Secreted Bone Biomarkers

Multi-analyte profiling of the protein levels in the cell culture medium of primary human osteoblasts was performed on the Luminex 200 system (Luminex, Austin, TX, USA), using xMAP technology. Acquired fluorescence data were analysed by the xPONENT 3.1 software (Luminex). The effect of adiponectin on the secretion of bone markers in the culture medium was measured using the Milliplex Human. The effects of RPM on the secretion of bone markers in the culture medium was measured using the Milliplex Human Bone Panel (HBNMAG-51K, MILLIPORE Corporation, Billerica, MA, USA). The assay included analysis of the following secreted biomarkers; adrenocorticotrophic hormone (ACTH), Dickkopf-1 (Dkk1), interleukin-6 (IL-6), insulin, leptin, tumour necrosis factor alpha (TNF-α), osteoprotegerin (OPG), osteocalcin (OC), osteopontin (SPP1), sclerostin (SOST), parathyroid hormone (PTH) and fibroblast growth factor-23 (FGF-23). Multi-analyte profiling of the protein levels of these factors in the cell culture medium was performed using the Luminex 200 system (Luminex, Austin, TX, USA), using xMAP technology. Acquired fluorescence data were analysed by applying the Luminex xPONENT version 3.1.871 or MILLIPLEX™ Analyst version 5.1 software (Luminex).

### 4.9. Histological Staining

After the 7 days and 14-days RPM culture of the hFOB 1.19 cells, the MCS were collected, three times carefully washed in PBS and fixed in 4% paraformaldehyde. After paraffin embedding, the MCS were sectioned with a microtome into 3 µm sections. Haematoxylin and eosin stains were applied, as described earlier [22]. Sirius red staining was performed to evaluate qualitative collagen content, as described earlier [87]. All sections were visualized by light microscopy, using an oil-immersion objective with calibrated magnifications of ×100 and ×200.

### 4.10. Determination of Calcification of the MCS

The multicellular spheroids were stained with the von Kossa silver staining technique [88]. The von Kossa stain was used to quantify mineralization in the 3D constructs. In addition, the MCS were stained with Alcian blue [89] (Sigma, Taufkirchen, Germany) and, subsequently, examined. For this purpose, the MCS were fixed, for 10 min, with 4% paraformaldehyde (PFA in DPBS), washed twice with DPBS, stained with a drop of the diluted staining solution (diluted according to the manufacturer’s instructions), washed twice with DPBS again and mounted with Entellan® (Merck, Darmstadt, Germany) for further microscopic analysis.

### 4.11. Pathway Analysis

To investigate the mutual regulation between genes and to visualize the localization and interactions between proteins, we entered the relevant UniProtKB entry numbers in the Pathway Studio v.11 software (Elsevier Research Solutions, Amsterdam, The Netherlands). Graphs were generated for gene expression, protein regulation and binding. The method was described previously [22]. 

### 4.12. Statistical Analysis 

All statistical analyses were performed using the SPSS 24.0 software (SPSS, Inc., Chicago, IL, USA). Data are presented as the mean ± SD. Due to the number of samples per group, an estimation of the data distribution characteristics was not feasible. We, therefore, employed the assumption-less, non-parametric Mann-Whitney-U test to assess the differences between the experimental groups. *p* < 0.05 was considered statistically significant.

## 5. Conclusions

The hFOB 1.19 cell line has been described as a model cell line for osteogenesis, with an ability to form mineralized nodules [37]. However, not all studies using the hFOB 1.19 cell line have published histological investigations [38,90]. Our histological staining identified cell distribution, including alterations of the cytoskeleton and the extracellular matrix and mineralization in the 3D constructs, having had up to 14 days of RPM exposure.

The human foetal osteoblasts exposed to the RPM grew as adherent monolayer, as well as in the form of RPM-derived bone constructs. We have successfully demonstrated the ability of the modelled microgravity analogue RPM to form 3D constructs and to support human osteogenesis in vitro. Given these findings, we conclude that hFOB 1.19 constructs show an excellent potential as biomaterial for human bone-tissue engineering applications.

## Figures and Tables

**Figure 1 ijms-20-01357-f001:**
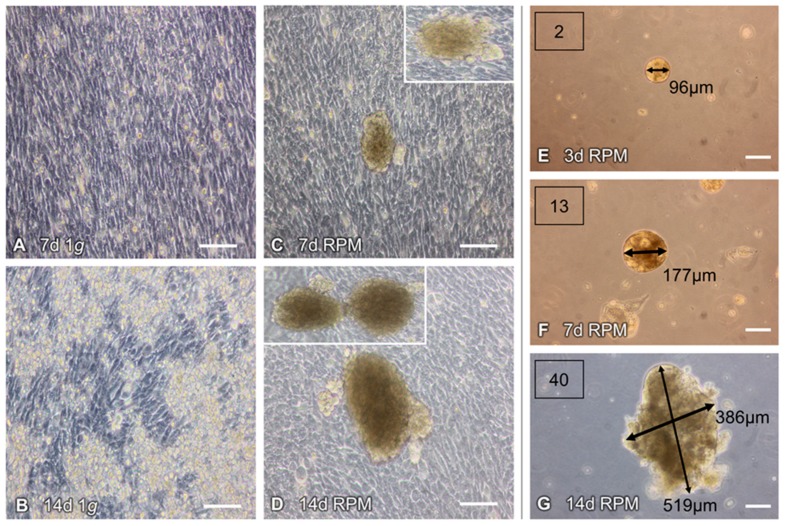
Random Positioning Machine (RPM)-induced formation of 3D multicellular spheroids (MCS) of human foetal osteoblasts (hFOB 1.19): Under static 1 *g* conditions, the cells grew adherently and remained in confluent monolayer cultures, when incubated for (**A**) 7 days and (**B**) 14 days. (**C**,**D**) Exposed to an RPM, the cells assembled into 3D aggregates, with the MCS increasing, over time, in size and number. (**E**) Microscopic analyses showed 2 MCS in the field of view, with an average diameter of 96 µm, after 3 days. After 7 days (**F**) and 14 days (**G**) cultivation on the RPM, the numbers of the spheroids increased from 13 to 40 MCS per field of view. The spheroids were 386 × 519 µm tall, after 14 days. The white squares in (**C**) and (**D**) show other examples of MCS. Scale bars: 100 µm; d, days.

**Figure 2 ijms-20-01357-f002:**
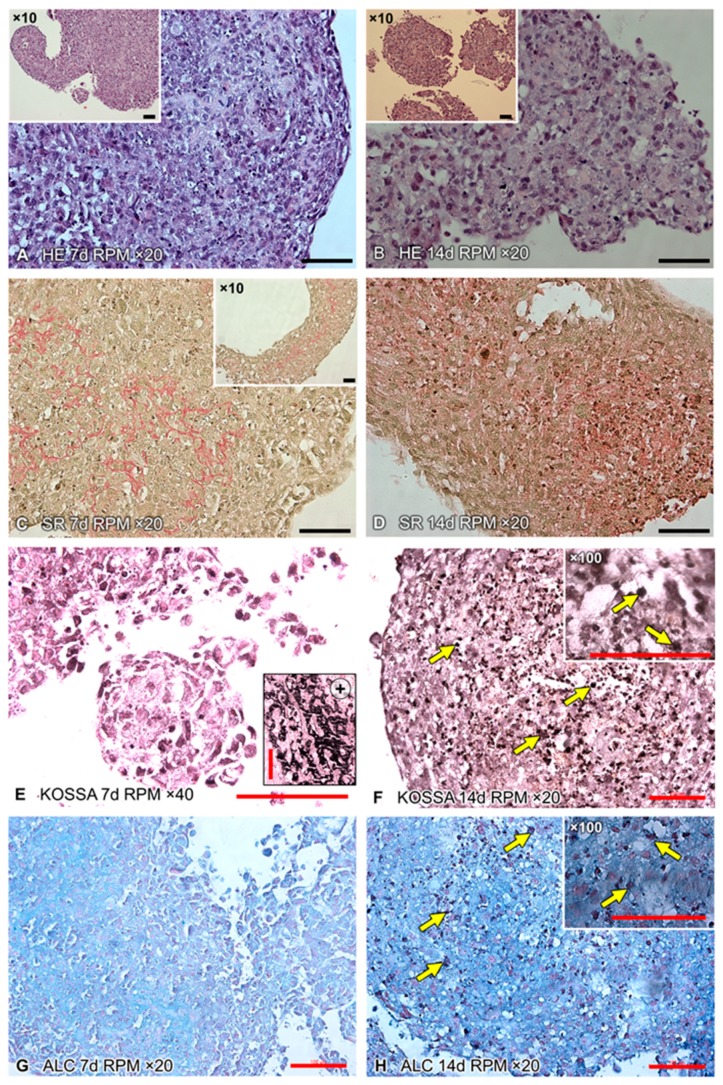
Histological assessment of paraffin-embedded cross-sections. The hFOB 1.19 cells were cultured for 7 days and 14 days on the RPM. 3D constructs were collected and, subsequently, paraffin-embedded. (**A**,**B**) HE staining shows the substantial aggregation of cells after 7 days and 14 days. (**C**,**D**) Sirius red-stained samples reveal a substantial collagen deposition. A clear increase is visible after 14 days, compared to 7 days. The black arrows indicate collagen deposition (red). Two stains were used to analyse biological mineralization within the MCS. (**E**,**F**) The von Kossa stain (KOSSA) is widely used for the detection of calcinosis in tissues. The small picture in (**E**) shows a positive control. (**G**,**H**) The Alcian blue (ALC) dye labels anionic tissue components, such as hyaluronate or chondroitin sulphate. The yellow arrows in (**F**) and (**H**) indicate regions of early calcification after 14 days. Scale bars: 100 µm, d, days.

**Figure 3 ijms-20-01357-f003:**
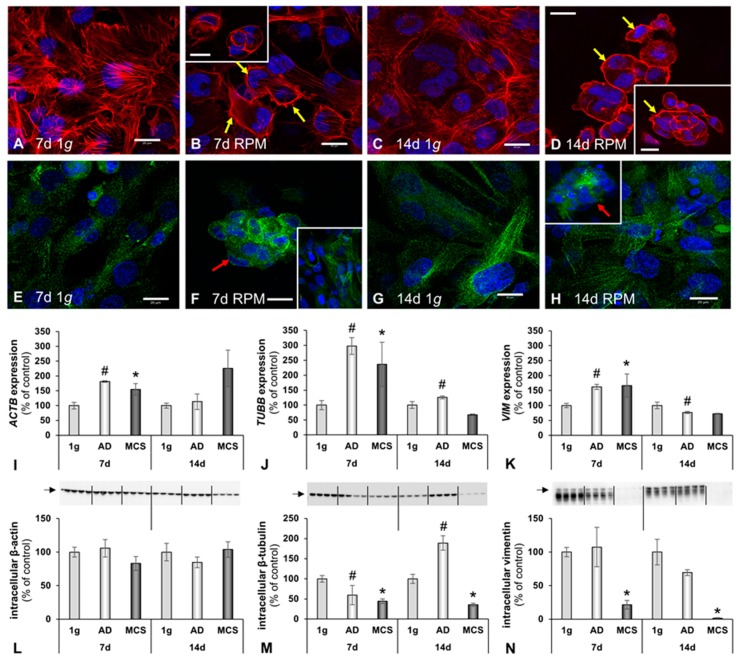
Influence of simulated microgravity on the cytoskeleton of human osteoblasts. Phalloidin–rhodamine staining visualized the F-actin distribution in the 1 *g* control cells (**A**,**C**) and RPM-exposed cells (**B**,**D**). The yellow arrows indicate filopodia and lamellipodia, as well as the accumulation of F-actin at the outer cellular membranes. (**E**–**H**) Tubulin immunofluorescence: Red arrows indicate holes in the tubulin network in spheroids. Gene expression of selected cytoskeletal genes: (**I**) β-actin (*ACTB*), (**J**) β-tubulin (*TUBB*) and (**K**) vimentin (*VIM*). Intracellular protein levels. (**L**) β-actin, (**M**) β-tubulin and (**N**) vimentin. All values are given as mean ± standard deviation. # *p* < 0.05 1 *g* vs. AD; * *p* < 0.05 1 *g* vs. MCS. Scale bars: 20 μm, d, days.

**Figure 4 ijms-20-01357-f004:**
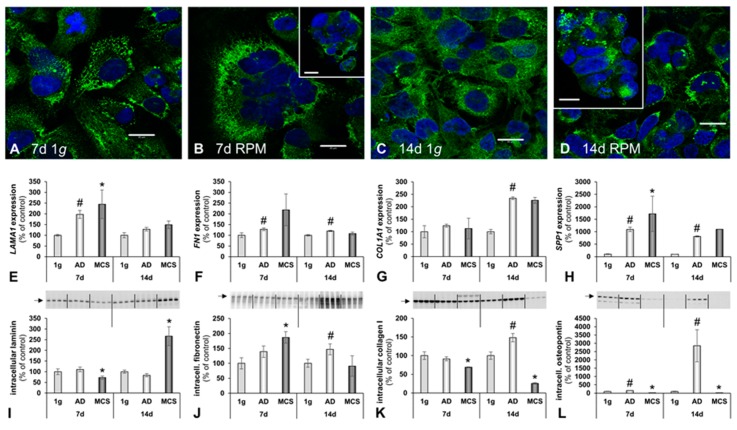
The influence of simulated microgravity on the extracellular matrix (ECM) of human osteoblasts. Laminin immunofluorescence. (**A**) 7 days 1 *g* control cells, (**B**) 7 days RPM exposed cells, (**C**) 14 days 1 g control cells and (**D**) 14 days RPM exposed cells. Gene expression of selected ECM genes: (**E**) Laminin α1 (*LAMA1*); (**F**) fibronectin (*FN1*); (**G**) collagen I subunit α1 (*COL1A1*); and (**H**) osteopontin (*SPP1*). Intracellular ECM protein levels: (**I**) Laminin; (**J**) fibronectin; (**K**) collagen type I; and (**L**) osteopontin. All values are given as mean ± standard deviation. # *p* < 0.05 1 g vs. AD; * *p* < 0.05 1 *g* vs. MCS. Scale bars: 20 μm, d, days.

**Figure 5 ijms-20-01357-f005:**
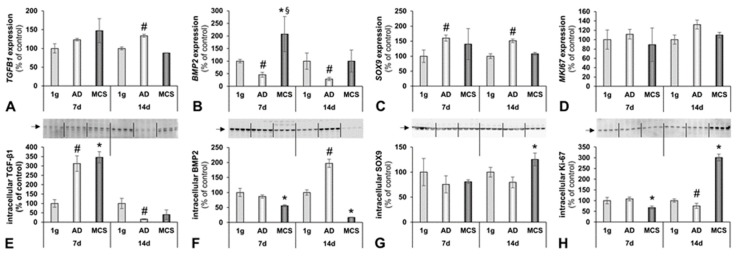
Bone-specific factors influenced by simulated microgravity. Gene expression of selected genes: (**A**) TGF-β_1_ (*TGFB1*); (**B**) BMP2 (*BMP2*); (**C**) SOX9 (*SOX9*); and (**D**) Ki-67 (*MKI67*). Intracellular ECM protein levels: (**E**) TGF-β1; (**F**) BMP2; (**G**) SOX9; and (**H**) Ki-67. All values are given as mean ± standard deviation. # *p* < 0.05 1 *g* vs. AD; * *p* < 0.05 1 *g* vs. MCS; § AD vs. MCS, *p* < 0.05, d, days.

**Figure 6 ijms-20-01357-f006:**
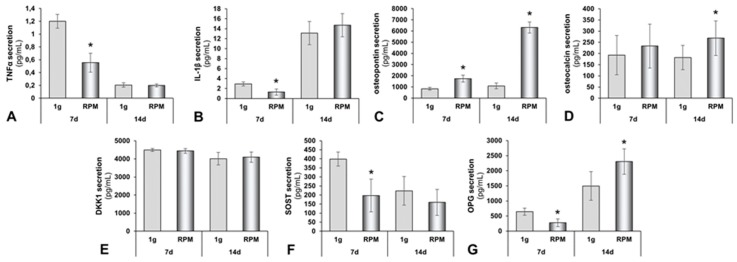
Protein secretion of human osteoblasts exposed to the RPM. (**A**) Tumour necrosis factor α (TNF-α); (**B**) interleukin-1β (IL-1β); (**C**) osteopontin; (**D**) osteocalcin; (**E**) dickkopf-1 (DKK1); (**F**) sclerostin (SOST); and (**G**) osteoprotegerin (OPG). All values are given as mean ± standard deviation. * *p* < 0.05 1 *g* vs. RPM, d, days.

**Figure 7 ijms-20-01357-f007:**
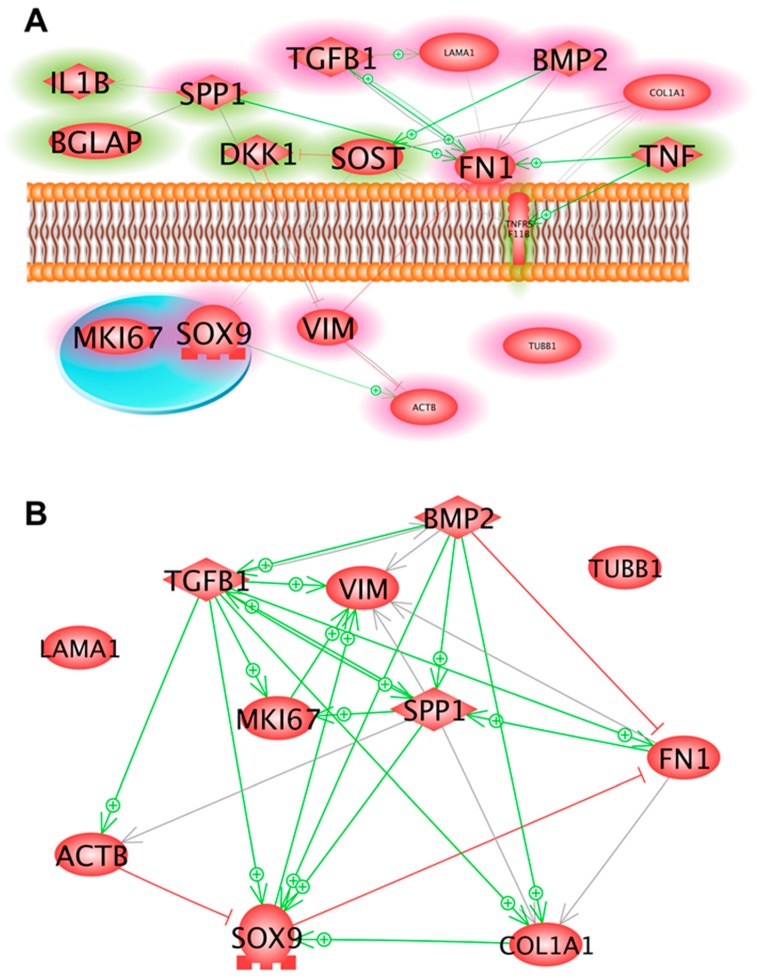
(**A**) Interaction and localization of proteins detected either in the supernatant by Luminex technology (green rims) or within the cells by Western blotting (red rims). Solid lines indicate binding, solid arrows show directed interaction and dashed arrows show influence. + signs point to an activity enhancing effect and red lines indicate inhibition. (**B**) Mutual interactions of the expression of the genes, investigated by qRT-PCR and shown in Figure 3I–K, Figure 4E–H, Figure 5A–D and in Table 1. + signs point to an activity-enhancing effect and red lines indicate inhibition. Each interaction network was built using Pathway Studio v.11.

**Figure 8 ijms-20-01357-f008:**
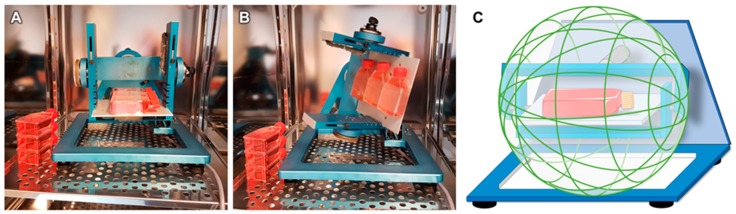
Experimental setup with the Random Positioning Machine (RPM). (**A**) The desktop-RPM (developed by AIRBUS Defence & Space, former: Dutch Space) is placed inside a standard incubator and was loaded with cells in bubble-free filled cell culture flasks. (**B**) After starting, it simulates microgravity by random rotation around two axes. The 1 g controls were placed next to the RPM. (**C**) The working principle of the RPM in real random mode. Parts of the figure were drawn by using pictures from Servier Medical Art. Servier Medical Art by Servier is licensed under a Creative Commons Attribution 3.0 Unported License (https://creativecommons.org/licenses/by/3.0/).

**Table 1 ijms-20-01357-t001:** Changes in gene expression, protein levels and secretion behaviour.

**mRNA Levels [% of 1 *g* Control]**																			
		**Gene**																	
**Sample**		***ACTB***	***BMP2***		***COL1A1***	***FN1***		***SPP1***		***LAMA1***	***MKI67***		***SOX9***		***TGFB1***	***TUBB***		***VIM***	
**7 days**	**1 *g***	100.0 ± 11.0	100.0 ± 6.7		100.0 ± 24.3	100.0 ± 10.8		100.0 ± 14.8		100.0 ± 3.6	100.0 ± 20.6		100.0 ± 20.8		100.0 ± 12.1	100.0 ± 15.1		100.0 ± 7.2	
	**AD**	**▲181.2** **± 2.9**	**▼45.6** **± 9.7**		124.0 ± 6.0	**▲127.8** **± 6.2**		**▲1094.0** **± 89.0**		**▲197.3** **± 18.4**	111.4 ± 10.3		**▲160.1** **± 10.1**		123.2 ± 3.5	**▲297.4** **± 27.8**		**▲162.3** **± 8.5**	
	**MCS**	**▲154.7** **± 19.7**	**▲207.2** **± 69.9**		112.6 ± 41.7	218.5 ± 73.9		**▲1719.2** **± 709.5**		**▲244.4** **± 67.0**	89.0 ± 35.9		140.1 ± 51.6		147.1 ± 32.3	**▲236.2** **± 74.5**		**▲166.7** **± 38.5**	
**14 days**	**1 *g***	100.0 ± 8.6	100.0 ± 32.1		100.0 ± 8.6	100.0 ± 3.6		100.0 ± 3.4		100.0 ± 11.5	100.0 ± 9.6		100.0 ± 8.1		100.0 ± 3.8	100.0 ± 11.3		100.0 ± 11.1	
	**AD**	113.1 ± 25.8	**▼28.8** **± 6.5**		**▲234.0** **± 5.3**	**▲119.8** **± 2.7**		**▲813.4** **± 22.2**		127.9 ± 8.1	132.1 ± 9.5		**▲151.2** **± 5.9**		**▲133.8** **± 4.0**	**▲125.7** **± 4.8**		**▼76.1** **± 3.5**	
	**MCS**	225.2 ± 61.7	100.1 ± 43.8		225.7 ± 11.4	107.7 ± 7.0		1098.2 ± 11.3		148.9 ± 17.6	109.7 ± 5.8		108.0 ± 4.3		87.7 ± 0.7	67.3 ± 2.7		72.1 ± 1.7	
**Intracellular Protein Levels [% of 1 *g* Control]**																			
		**Protein**																	
**Sample**		**β-actin**	**BMP2**		**Col 1**	**FN**		**OPN**		**laminin**	**Ki-67**		**SOX9**		**TGF-β1**	**β-tubulin**		**vimentin**	
**7 days**	**1 *g***	100.0 ± 7.5	100.0 ± 13.6		100.0 ± 10.4	100.0 ± 18.8		100.0 ± 16.6		100.0 ± 13.2	100.0 ± 14.8		100.0 ± 27.3		100.0 ± 20.1	100.0 ± 8.2		100.0 ± 6.9	
	**AD**	105.8 ± 12.9	86.5 ± 5.4		91.4 ± 5.2	139.2 ± 19.2		**▲154.4** **± 13.2**		110.7 ± 11.1	107.8 ± 9.1		75.3 ± 17.0		**▲312.1** **± 41.4**	**▼59.7** **± 23.9**		107.4 ± 29.3	
	**MCS**	83.2 ± 10.4	**▼55.7** **± 3.2**		**▼68.8** **± 1.1**	**▲186.9** **± 19.1**		**▼19.4** **± 7.5**		**▼72.1** **± 7.7**	**▼67.5** **± 6.5**		80.9 ± 3.7		**▲345.9** **± 30.4**	**▼44.3** **± 5.6**		**▼21.5** **± 6.3**	
**14 days**	**1 *g***	100.0 ± 13.1	100.0 ± 8.8		100.0 ± 9.8	100.0 ± 13.2		100.0 ± 24.6		100.0 ± 7.6	100.0 ± 7.5		100.0 ± 9.6		100.0 ± 27.3	100.0 ± 11.3		100.0 ± 19.2	
	**AD**	84.8 ± 8.0	**▲196.9** **± 13.6**		**▲148.1** **± 11.1**	**▲146.9** **± 18.0**		**▲2852.2** **± 964.4**		83.5 ± 7.6	**▼75.3** **± 12.0**		79.6 ± 10.8		**▼15.5** **± 2.4**	**▲188.8** **± 18.0**		69.3 ± 4.3	
	**MCS**	103.9 ± 11.7	**▼16.9** **± 1.0**		**▼25.5** **± 2.1**	**91.0** **± 34.3**		**▼14.8** **± 17.3**		**▲266.4** **± 44.4**	**▲300.8** **± 15.9**		**▲125.4** **± 12.8**		39.8 ± 26.0	**▼35.7** **± 3.8**		**▼1.6** **± 0.8**	
**Protein Secretion [% of 1 *g* Control]**																			
		**Protein**																	
**Sample**		**DKK1**		**IL-1β**			**OC**		**OPG**			**OPN**		**SOST**			**TNF-α**		
**7 days**	**1 *g***	100.0 ± 1.4		100.0 ± 13.0			100.0 ± 32.6		100.0 ± 15.3			100.0 ± 14.7		100.0 ± 7.8			100.0 ± 8.7		
	**RPM**	98.8 ± 2.5		**▼34.8** **± 5.7**			112.0 ± 47.8		**▼42.6** **± 19.0**			**▲206.7** **± 38.1**		**▼49.4** **± 16.9**			**▼46.2** **± 12.2**		
**14 days**	**1 *g***	100.0 ± 6.6		100.0 ± 11.2			100.0 ± 24.9		100.0 ± 26.2			100.0 ± 14.8		100.0 ± 32.6			100.0 ± 7.5		
	**RPM**	102.1 ± 6.0		125.6 ± 13.5			**▲148.9** **± 28.0**		**▲154.1** **± 22.5**			**▲579.2** **± 20.8**		74.4 ± 21.4			96.0 ± 11.0		

▲ significant increase/up-regulation (*p* < 0.05); ▼ significant decrease/down-regulation (*p* < 0.05).

**Table 2 ijms-20-01357-t002:** Primers used for qRT-PCR.

Gene	Primer Name	Sequence
*18S rRNA*	18S-F	GGAGCCTGCGGCTTAATTT
	18S-R	CAACTAAGAACGGCCATGCA
*ACTB*	ACTB-F	TGCCGACAGGATGCAGAAG
	ACTB-R	GCCGATCCACACGGAGTACT
*BMP2*	BMP2-F	GACCTGTATCGCAGGCACTCA
	BMP2-R	TCGTTTCTGGTAGTTCTTCCAAAGA
*COL1A1*	COL1A1-F	ACGAAGACATCCCACCAATCAC
	COL1A1-R	CGTTGTCGCAGACGCAGAT
*FN1*	FN1-F	TGAGGAGCATGGTTTTAGGAGAA
	FN1-R	TCCTCATTTACATTCGGCGTATAC
*LAMA1*	LAMA1-F	TGCTCATGGTCAATGCTAATCTG
	LAMA1-R	TCTATCAATCCTCTTCCTTGGACAA
*MKI67*	MKI67-F	TGGGGAAAGTAGGTGTGAAAGAAG
	MKI67-R	CTCCTTAAACGTTCTGATGCTCTTG
*SOX9*	SOX9-F	TTGAGACCTTCGACGTCAATGAG
	SOX9-R	TCTGGCCACGAGTGGCC
*SPP1*	SPP1-F	CGAGGTGATAGTGTGGTTTATGGA
	SPP1-R	CGTCTGTAGCATCAGGGTACTG
*TBP*	TATA-F	GTGACCCAGCATCACTGTTTC
	TATA-R	GCAAACCAGAAACCCTTGCG
*TGFB1*	TGFB1-F	CACCCGCGTGCTAATGGT
	TGFB1-R	AGAGCAACACGGGTTCAGGTA
*TUBB*	TUBB-F	CTGGACCGCATCTCTGTGTACTAC
	TUBB-R	GACCTGAGCGAACAGAGTCCAT
*VIM*	VIM-F	TTCAGAGAGAGGAAGCCGAAAAC
	VIM-R	AGATTCCACTTTGCGTTCAAGGT

All sequences are given in 5′–3′ direction.

**Table 3 ijms-20-01357-t003:** Antibodies used for Western blot analyses.

Antibody Target	Company/No	Host	Size (kDa)	Used Dilution
β-actin	Sigma/A5316	Ms	42	1:2000
BMP2	Novus biological/	Rb	44	1:500
Collagen I	Sigma/C2456	Ms	~130	1:500
Cyclophilin	Abcam/AB178397	Rb	20	1:2000
Fibronectin	Invitrogen/MA1198	Ms	250	1:1000
Ki-67	Santa cruz/c-15402	Rb	250	1:200
Laminin	Invitrogen/PA1-1673	Rb	~200	1:1000
Osteopontin	Sigma/07264	Rb	50	1:1000
SOX9	Abcam/ab26414	Rb	56	1:1000
TGF-β1	Abcam/ab92486	Rb	44	1:500
TNF-α	Abcam/ab9739	Rb	26	1:500
β-tubulin	Sigma/T5293	Ms	50	1:1000
Vimentin	Sigma/WH0007431	Ms	~53	1:1000

Ms: mouse, Rb: rabbit.

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
