# Peer review of "Changes in Human Foetal Osteoblasts Exposed to the Random Positioning Machine and Bone Construct Tissue Engineering"

_ijms, 2019, doi:10.3390/ijms20061357_

Reviewer 1 Report

The growing amount of space missions and the growing interest to colonize distant planets, increased research studies on the effects induced by the microgravity environment on living beings. In particular, experimental models, even if with some limits, are useful to this aim but also to study responses at cellular levels. This is the case of the study reported in this manuscript in the field of bone physiology. The results that the Authors obtained, fit well into the panorama of the knowledge regarding this field, but some points have to be better elucidated. A list of these points as they appear in the manuscript, is reported below.

1- Abstract. The Authors reported: “….the purpose to engineer 3D bone constructs.”, and listed their results. This makes the abstract quite inconsistent with the topic of the special issue, and lacks of concluding remarks.

2- The cellular model subjected to RPM, was monitored at 7 and 14 days. The Authors have to explain the reason of this time choice.

3- In figure 2, the Authors showed 20x and 10x magnifications, but it is no clearly evident for each panel.

4- The Authors reported mRNA and protein expression as “% of control”, what did they mean?  They referred to samples at static conditions? Or to a housekeeping gene or protein as loading control? The last one is important for the quantitative analyses avoiding misinterpretations.

5- The Discussion is extremely analytic, but in some points appeared too long repeating the description of the results. I noticed that after the section of Materials and Methods there is the section of Conclusions, supposing it was required by the journal guidelines. On the other hand, a reader can find the Discussion lacking of a concluding summary that pointed out the relevance of the results reported in the manuscript, so it could be of help at least of a concluding sentence

6) Revise English style and grammar. For example: the sentence “Furthermore, under the microscope adjusted to a magnification of ×100, three times more MCS per field of view were seen on day 14 than on day 7 (numbers in squares are inserted in Fig. 1E,F,G).” at pag 3 is not clear; in the legend of figure 1 it should be reported: “…..Afetr 7d (F) and 14d (G) on the RPM….”; the same in the legend of figure 3: “in 1g-control cells (A, C) and RPM exposed cells (B, D)…..”, please check all legends.

    Author Response

The authors would like to thank the editors and the reviewers for their important suggestions. All points were addressed and the changes highlighted in yellow in the revised manuscript. Below please find our answers to your questions and suggestions point-by-point.

Reviewer 1

 Comments and Suggestions for Authors

The growing amount of space missions and the growing interest to colonize distant planets, increased research studies on the effects induced by the microgravity environment on living beings. In particular, experimental models, even if with some limits, are useful to this aim but also to study responses at cellular levels. This is the case of the study reported in this manuscript in the field of bone physiology. The results that the Authors obtained, fit well into the panorama of the knowledge regarding this field, but some points have to be better elucidated. A list of these points as they appear in the manuscript, is reported below.

Abstract. The Authors reported: “….the purpose to engineer 3D bone constructs.”, and listed their results. This makes the abstract quite inconsistent with the topic of the special issue, and lacks of concluding remarks.

Answer: We have revised the objective of the study and added a concluding sentence to the abstract in the revised manuscript, please see lines 36-38.

2- The cellular model subjected to RPM, was monitored at 7 and 14 days. The Authors have to explain the reason of this time choice.

Answer: The time choice 7 and 14 days was used because of the expertise of Professor Sundaresan with bone tissue engineering technologies. Living bone constructs had been investigated earlier after a 7-day-culture on the Rotating Wall Vessel (please see ref. 36. Professor Grimm has a long-term expertise with the application of the RPM for tissue engineering purposes. 7 and 14 day-cultures had been suitable for cartilage and tubular construct tissue engineering (ref. 27, 29).

Furthermore, after 14 days mineralization is detectable in the bone constructs.

  3- In figure 2, the Authors showed 20x and 10x magnifications, but it is no clearly evident for each panel.

Answer: The magnifications are given in each picture. We have improved the visibility of the scale bars and the magnifications.

 4- The Authors reported mRNA and protein expression as “% of control”, what did they mean?  They referred to samples at static conditions? Or to a housekeeping gene or protein as loading control? The last one is important for the quantitative analyses avoiding misinterpretations.

Answer: Thank you for your comment. “% of control” means that the control experiment (static condition) is set as a 100% while the results are presented in relation to the controls. That makes changes easier to spot. We have clarified this in the methods part. Normalization of the data has been done using 18S rRNA and TBP in case of the qPCR and Cyclophilin in case of Western blots.

 5- The Discussion is extremely analytic, but in some points appeared too long repeating the description of the results. I noticed that after the section of Materials and Methods there is the section of Conclusions, supposing it was required by the journal guidelines. On the other hand, a reader can find the Discussion lacking of a concluding summary that pointed out the relevance of the results reported in the manuscript, so it could be of help at least of a concluding sentence

Answer: The Conclusions after the section Materials and Methods is placed according to  the journal guidelines. We have included a concluding summary at the end of the discussion.

 6) Revise English style and grammar. For example: the sentence “Furthermore, under the microscope adjusted to a magnification of ×100, three times more MCS per field of view were seen on day 14 than on day 7 (numbers in squares are inserted in Fig. 1E,F,G).” on page 3 is not clear; in the legend of figure 1 it should be reported: “…..Afetr 7d (F) and 14d (G) on the RPM….”; the same in the legend of figure 3: “in 1g-control cells (A, C) and RPM exposed cells (B, D)…..”, please check all legends.

Answer: We have corrected these examples. English style and grammar will be corrected by IJMS.

 Reviewer 2 Report

REE’S REPORT – COMMENTS TO THE AUTHORS

Mann et al.:

Paper ID: : ijms-455152

Paper Title: "Influence of simulated microgravity on growth of human fetal osteoblasts and characterization of the derived three dimensional bone constructs”

 This is a very well written, concise and comprehensive manuscript (MS) dealing with the effect of random positioning (a means regarded by some authors to model microgravity) on human fetal osteoblasts with the major outcome that random positioning yielded spheroids showing a bone-specific morphology and corresponding biomolecular markers.

Certainly, this finding is of great biomedical relevance.

 The MS should be published provided that the following minor points of concern are addressed in a minor revision of the MS:

 Minor points of criticism

Doubts have been raised, if random      positioning (RP) in fact is a simulation of microgravity, since RP      produces eminent shearing forces, which might fog any possible effect the      lack of sedimentation during RP might have. In a revised MS, the authors      should, in their introduction, note that a random positioning machine      (RPM) is regarded a microgravity simulator by some authors, while numerous      publications suggest that it is not. Therefore, the authors should avoid      the term “microgravity simulation” all over the following text and      captions to figures, but rather use the term “random positioning”.

This is to avoid a potential reader without background knowledge to use an RPM for her/his samples producing hypergravity and/or shear forces rather than microgravity/weightlessness, seen as such from the perspective of the biosystem.

Hence, I would like to suggest to use the term “random positioning/simulated weightlessness” rather than “simulated microgravity” in the title of the MS.

If possible, please add to your discussion      at appropriate places in which way your specific results may aid in      medication (or the development of medical support) in the future. In other      words: Discuss your overall objective.

Caption of Fig. 1. It is not possible to      model a simulation.

Fig. 2: The scale bars are almost      invisible.

 Author Response

The authors would like to thank the editors and the reviewers for their important suggestions. All points were addressed and the changes highlighted in yellow in the revised manuscript. Below please find our answers to your questions and suggestions point-by-point.

Reviewer 2

Top of Form

Mann et al.:

Paper ID: : ijms-455152

Paper Title: "Influence of simulated microgravity on growth of human fetal osteoblasts and characterization of the derived three-dimensional bone constructs”

 This is a very well written, concise and comprehensive manuscript (MS) dealing with the effect of random positioning (a means regarded by some authors to model microgravity) on human fetal osteoblasts with the major outcome that random positioning yielded spheroids showing a bone-specific morphology and corresponding biomolecular markers.

Certainly, this finding is of great biomedical relevance.

 The MS should be published provided that the following minor points of concern are addressed in a minor revision of the MS:

 Minor points of criticism

Doubts have been raised, if random      positioning (RP) in fact is a simulation of microgravity, since RP      produces eminent shearing forces, which might fog any possible effect the      lack of sedimentation during RP might have. In a revised MS, the authors      should, in their introduction, note that a random positioning machine      (RPM) is regarded a microgravity simulator by some authors, while numerous      publications suggest that it is not. Therefore, the authors should avoid      the term “microgravity simulation” all over the following text and      captions to figures, but rather use the term “random positioning”.

Answer: We have addressed this point in the introduction, please see page 2, lines 50-62.

This is to avoid a potential reader without background knowledge to use an RPM for her/his samples producing hypergravity and/or shear forces rather than microgravity/weightlessness, seen as such from the perspective of the biosystem.

 Answer: We have included additional references 5, 6 and 8.

 Hence, I would like to suggest to use the term “random positioning/simulated weightlessness” rather than “simulated microgravity” in the title of the MS.

Answer: We have changed the title accordingly.

If possible, please add to your discussion      at appropriate places in which way your specific results may aid in medication (or the development of medical support) in the future. In other words: Discuss your overall objective.

Answer: Thank you for this suggestion. We have discussed this point, please see lines 454-465.

Caption of Fig. 1. It is not possible to      model a simulation.

Answer: We have changed this error.

Fig. 2: The scale bars are almost      invisible.

Answer: We have improved the visibility of the scale bars.